# Thailand Prevalence and Profile of Food Insecurity in Households with under Five Years Children: Analysis of 2019 Multi-Cluster Indicator Survey

**DOI:** 10.3390/ijerph19095065

**Published:** 2022-04-21

**Authors:** Jintana Jankhotkaew, Orana Chandrasiri, Sorasak Charoensit, Vuthiphan Vongmongkol, Viroj Tangcharoensathien

**Affiliations:** International Health Policy Program, Ministry of Public Health, Tiwanon Rd., Nonthaburi 11000, Thailand; jintana@ihpp.thaigov.net (J.J.); sorasak0063@gmail.com (S.C.); vuthiphan@ihpp.thaigov.net (V.V.); viroj@ihpp.thaigov.net (V.T.)

**Keywords:** food insecurity, socio-economic status, children under five, households, Thailand, food insecurity experience scale

## Abstract

This study aimed to estimate the prevalence and profile of food insecurity in households with children under 5 years old using the Food Insecurity Experience Scale (FIES) in Thailand. We integrated FIES into the 2019 Multiple Indicator Cluster Surveys (MICS). A total of 861 households were successfully interviewed with FIES. The Rasch model was applied to examine the validity and reliability. Multiple logistic regression was used to assess the association between socio-economic status and prevalence of food insecurity, adjusting for geographical regions and characteristics of households. We found that FIES measurement is valid as Infit falls within the normal range of 0.7–1.3 and is reliable (Rasch reliability value of 0.81). The overall prevalence of moderate or severe food insecurity was 2.79%. The wealthiest households were less likely to suffer from food insecurity than the poorest households (adjusted OR: 0.07; 95% CI: 0.02–0.34; *p*-value < 0.05). Households with children under 5 years old living in rural areas had lower food insecurity severity scores. We recommend social protection policies such as food and nutrition subsidies or conditional cash transfer to poor households with children under the age of 5.

## 1. Introduction

One-fourth of the world’s population suffers from food insecurity. The situation is more severe in the African region, accounting for more than half of the population in the region. In 2019, it was estimated that 2 billion people worldwide (26% out of the world population of 7.7 billion) suffered from food insecurity at moderate or severe levels [1].

Food insecurity has a negative impact on the health of children and young people and their development in later life. It increases the risk of stunting and wasting among children [2] and causes mental health problems among adults [3]. It is also a threat to human development, particularly early childhood development [4,5]. Food insecurity exacerbates children’s physical and mental development, which later impacts their health and their capacity to achieve the highest potential in their lives [6]. In addition, malnutrition is a key risk factor for non-communicable diseases (NCDs). Exposure to early under-nutrition followed by later overweight increases the risk of non-communicable diseases [7].

Globally, food insecurity has been recognized as a threat to health. Food insecurity has been agreed and committed by head of states of all United Nations Member States in the Sustainable Development Goals (SDG), target 2.1 (end hunger and ensure access by all people, in particular the poor and people in vulnerable situations, including infants, to safe, nutritious, and sufficient food all year round), as measured by SDG indicator 2.1.1 (prevalence of undernourishment) and indicator 2.1.2 (prevalence of moderate or severe food insecurity in the population, based on the Food Insecurity Experience Scale (FIES)). Further, there are other relevant SDG targets related to food insecurity, including target 2.2.1 prevalence of stunting (height for age <−2 standard deviation from the median of the World Health Organization (WHO) Child Growth Standards) among children under 5 years of age. Undernutrition has health risks to children, such as premature death from common infections, delayed recovery, and learning capabilities [7].

Importantly, reducing food insecurity requires a commitment from various sectors at national and global levels [8]. It also requires evidence to effectively inform governments on the prevalence and trend of both food insecurity and malnutrition as well as identifying risk groups to ensure equitable access to effective interventions.

Households with less ability to access resources suffer most from food insecurity. Studies investigating the association between socioeconomic status (SES) are often done in countries where food insecurity is more prevalent such as Bangladesh (48%) [9], Cambodia (82%) [10], Ethiopia (75%) [11,12], India (77%) [13], and Nepal (54%) [14]. On the other hand, a study in Australia, where food insecurity is less prevalent, found that children living in families with financial support from the government were more likely to experience food insecurity. The same study reported that children living in medium disadvantaged areas were more likely to experience food insecurity than in the most disadvantaged areas [15]. However, the study applied community-level SES, which may not reflect households’ SES. Moreover, indicators for measuring SES vary across studies, including income, education, occupation, and household asset index. This study applied the most comprehensive tool, using household assets recommended by the World Bank, and it suits the context of low-and middle-income countries [16].

In Thailand, malnutrition is still a problem, with a prevalence of stunting that increased from 10.5% in 2016 to 13.3% in 2019, and the prevalence of wasting increased from 5.4% to 7.7% during the same period [17,18]. There has been no study done to investigate the prevalence of food insecurity at the national level; food insecurity is one of the risk factors of stunting.

Although the Food and Agriculture Organization of the United Nations (FAO) reported a low prevalence of moderate or severe food insecurity (4.8%) and severe food insecurity (0.5%) in Thailand [19], it did not investigate food insecurity across socio-demographic profiles. Another study conducted in ethnic communities in the Northern region assessed the food insecurity in households with children under 5 years old, by using the Household Food Insecurity Access Prevalence (HFIAP) measurement. It reported that 74% of sample households suffered from food insecurity. However, this study was conducted in small areas with limited numbers of sample size (172 households) [20].

This study fills the knowledge gaps. First, it is one among few studies that investigate the association between SES and food insecurity prevalence and level of severity in a low-prevalence food insecurity context. Second, it gains policy attention; as otherwise, the vulnerable households in low SES are left behind. Children under 5 years old need adequate food and nutrition for their cognitive and physical development. Third, though FIES has been applied in Thailand by Gallup World Poll; it does not provide evidence on the validity and reliability of the FIES tool. The validation of the tool is required in order to apply to a national scale survey, as required by SDG indicator 2.1.2 “Prevalence of moderate or severe food insecurity in the population, based on the Food Insecurity Experience Scale”.

This study aims to estimate the prevalence of food insecurity in households with children under 5 years old and analyze the association between socio-economic status and food insecurity.

## 2. Materials and Methods

### 2.1. Study Design and Data Collection

In consultation with the National Statistical Office of Thailand (NSO), we integrated the Food Insecurity Experience Scale (FIES) module into the 2019 Thailand Multiple Indicator Cluster Survey round six (MICS6).

MICS6 applied a stratified two-stage sampling technique. Urban and rural enumeration areas were systematic random samples using a probability proportional to size in all 77 provinces. In each enumeration area, 10 households with and without children under 5 years old were randomly selected. The total sample households for MICS6 was 40,660. See details of the study design elsewhere [21].

For the FIES module, a subsample of MICS6 households with children under 5 was used. We calculated the sample household based on 4.8% prevalence of food insecurity in Thailand, using a formula “*n* = [DEFF × Np(1−*p*)]/[(d^2^/Z^2^_1−α/2_ × (N − 1) + *p* × (1−*p*)]” in the OpenEpi program [22]; N = 3,584,000 [23]; design effect = 3; confident interval = 95%; and prevalence = 4.8%. The sample household was 1317; after adjusting for response rates of 99.5% [18], the samples were 1331 households. In consultation with NSO, a total of 1500 households were agreed. This was consistent with the sample size recommended by the FAO of at least 1000 households [19]. Finally, NSO provided 16 provinces randomly selected from four regions (north, northeast, central, and south) for the FIES survey.

However, NSO successfully interviewed 861 households using FIES. The low response rate, 57.4%, was due to the outdated household sampling frame. For example, children moved out of the households, or they were older than 5 years. Only 2% of households refused to participate in the FIES interview.

The person who prepared food for family members was interviewed with FIES. Trained staffs at NSO provincial office conducted face-to-face interviews in November 2019.

### 2.2. Measurements

FIES, a standardized tool developed by FAO, was translated into Thai by International Health Policy Program (IHPP) and NSO through consultative meetings with experts, ensuring it fits with the Thai context. It was pre-tested along with MICS6 survey questions. IHPP team members attended a FAO training course (19–23 November 2018) on FIES principle, analytical technique and policy utility.

#### 2.2.1. Outcome

This study assessed food insecurity at the household level by asking persons who prepared food for the households to respond to eight standardized questions in the FIES. Respondents were asked their experience, with reference to the last 12-month recall period: (1) Was there a time when you (or any individuals in the household) were worried you would not have enough food to eat because of a lack of money or other resources? (2) Was there a time when you (or any individuals in the household) were unable to eat healthy and nutritious food because of a lack of money or other resources? (3) Was there a time when you (or any individuals in the household) ate only a few kinds of foods because of a lack of money or other resources? (4) Was there a time when you (or any individuals in the household) had to skip a meal because there was not enough money or other resources to get food? (5) Was there a time when you (or any individuals in the household) ate less than you thought you should because of a lack of money or other resources? (6) Was there a time when your household ran out of food because of a lack of money or other resources? (7) Was there a time when you (or any individuals in the household) were hungry but did not eat because there was not enough money or other resources for food? (8) Was there a time when you (or any individuals in the household) went without eating for a whole day because of a lack of money or other resources? The answers are Yes = 1 score, No = 0 score, and Don’t know.

#### 2.2.2. Main Independent Variables: Household Asset Index

To construct a household asset index, we assessed the ownership of the household durables; a set of housing characteristics and household assets were asked to the head of households; see the list of household assets in the MICS6 report [21]. Respondents answered “yes” or “no”. We used principal component analysis to generate factor scores and later categorized these into a tertile wealth index. The first tertile represents the poorest 33.3% of households, and the third tertile represents the richest households. Due to the low prevalence of food insecurity, we did not categorize the wealth index into quintiles.

#### 2.2.3. Covariates

We collected variables based on the context of the country and on previous studies [10,12,15]. These variables included the source of food for the households (purchased, homegrown), regional geography, urban or rural, number of household numbers, and whether the household received financial support from the Thai government.

### 2.3. Data Analysis

#### 2.3.1. Estimating Food Insecurity

The Rasch model was used for FIES analysis as recommended by the FAO. The model examines the validity and reliability of FIES across different country contexts [19]. We applied four tests to assess the validity and reliability [24].

First, the infit test, with a normal range of 0.7–1.3, was used to assess the validity of the FIES. Second, the outfit test, which is more sensitive to unexpected response patterns, was used to measure the validity. These unusual patterns represented those who reported severe levels of food insecurity but did not report mild insecurity. For example, respondents who answered yes to the following question “went without eating for a whole day” but answered “no” to the following question “were you worried you would not have enough food to eat?”. The normal range of the outfit value is less than 2.

Third, the residual correlation matrix was used to measure the redundancy of the questions. The normal value is −4 to 4. Lastly, the Rasch reliability test was used to measure the reliability of the FIES. The normal value is greater than 0.8. After excluding extreme values (scores of 0 and 8) as recommended by the FAO, we had 148 cases with non-extreme scores for testing the validity and reliability. With reference to FAO guidelines, respondents who reported raw scores ranging from 1 to 7 were included in the test [19,24]. In our study, there were limitations in the interpretation of findings regarding the outfit test, the residual correlation matrix, and the Rasch reliability because there is a low number of cases with non-extreme scores (less than 300 cases) [24]. In this step, we consulted and received confirmation from the FAO regarding the accuracy of our analysis.

After testing the validity and reliability of the FIES, we categorized households into two groups: food-secure households (total raw score of 0–3) and food-insecure households (total raw score of 4–8) [25]. We combined moderate and severe food insecurity due to their low prevalence.

#### 2.3.2. Univariate and Multivariate Analysis

We used the chi-square test to demonstrate the differences in the annual prevalence of food insecurity across SES and demographic profiles.

We applied the two models for multiple regression. There are two main purposes for selecting the two models. First, we selected multiple logistic regression to determine the association between socio-economic status and food insecurity experience (binary outcome—did not experience/experienced). Second, we selected Tobit regression to determine the association between socio-economic status and the severity of food insecurity (raw score of food insecurity: 0–8).

Multiple logistic regression analysis was employed to investigate the association between SES and the prevalence of food insecurity. We adjusted for the following covariates: source of food, regional geographical area, place of residence (urban or rural), household size, and whether the household received financial support from the Thai government.

We also employed a Tobit regression model to investigate the association between SES and the severity of food insecurity (raw scores). The raw scores are the sum of scores given to the eight FIES questions; 0 means no food insecurity, and 8 represents the highest level of food insecurity. We used the Tobit regression model because the raw score was concentrated at 0. Therefore, we censored raw scores at 0. The model was adjusted with the same covariates as in the logistic regression model.

We also attempted to investigate the association between food insecurity and malnutrition among children under 5 years old. However, the small sample size and the low prevalence of food insecurity did not allow us to perform this analysis.

## 3. Results

### 3.1. Validity and Reliability of FIES

We found that the FIES was suited to the Thai context as the infit test was in the normal range. The outfit test resulted in two questions lying outside the normal range (Table 1). However, considering the small number of households, we considered the results of the infit test for validity. We found that the residual correlation matrix and the Rasch reliability value of 0.81 was within the normal range.

### 3.2. Characteristics of Households

Of 861 households, 24 households (2.79%) suffered from food insecurity. Of these, 22 (92%) and 2 (8%) reported moderate or severe food insecurity, respectively (Table 2).

The majority of the households purchased food, lived in the Northern region and rural areas, and received financial support from the Thai government. There were, on average, five members per household.

### 3.3. Association between Socio-Economic Status and Food Insecurity

Based on univariate analysis, the poorest household tertile had a greater prevalence of food insecurity than the middle and richest households (*p*-value < 0.05). Households in the southern region had the highest prevalence of food insecurity compared to other regions (*p*-value < 0.05) (Table 2).

On multivariate analysis, households in the middle (adjusted OR 0.11; 95% CI 0.03, 0.37; *p* < 0.05) and richest tertiles (adjusted OR 0.07; 95% CI 0.02, 0.34; *p* < 0.05) were less likely to suffer from food insecurity compared to the poorest households (Table 3). These results were supported by the Tobit regression model (Table 4).

Considering other covariates, households that lived in the northern region were less likely to suffer from food insecurity than households in the central region (Table 3). However, when considering the severity scores, households in the southern region had higher scores of food insecurity than households in the central region (coefficient 2.60; 95% CI 1.41, 3.78; *p*-value < 0.05). Additionally, households that relied on homegrown food had higher food insecurity scores than households that relied on purchasing their food (coefficient 2.01; 95% CI 0.91, 3.11; *p*-value < 0.05). Furthermore, we found that households in rural areas had lower food insecurity scores than households in urban areas (coefficient: −0.84; 95% CI: −1.58, −0.10; *p*-value < 0.05) (Table 4).

## 4. Discussions

The study found a low prevalence of food insecurity (2.79%) among households with children under 5 years old. We found that the FIES is a valid measure for assessing food insecurity. The study also found that the poorest household tertiles suffered higher levels of food insecurity than the richest household tertiles. Additionally, rural areas had lower food insecurity scores than households in urban areas.

To the best of our knowledge, this is the first national survey conducted in Thailand to investigate the prevalence of food insecurity in households with children under 5 years old. The prevalence was lower than the 2016 FAO estimate of 4.8% for moderate or severe food insecurity in all Thai households. However, our study (households with children under 5 years old) is not comparable to the FAO survey since people of all ages were included. One of the main reasons that the result was different was because of the sampling frame and sampling methods. This research applied a NSO sampling frame while the FAO report was collected from the Gallup World Poll [19]. When we compared the findings with a study conducted in Northern Thailand, the prevalence was lower (74%) [20]. However, the two studies have limitations in comparison because the study has different sampling areas as the previous study was conducted in ethnic communities, while our study was conducted across Thailand.

The study ensured the validity of the FIES questionnaires in the Thai context as the infit value laid in the normal range for all questions. As this is the first survey conducted in Thailand, the tool from this study can be applied in future studies to investigate the food insecurity situation and investigate the progress of interventions on food insecurity at national and local levels.

This study is one among few conducted in the Southeast Asia Region and experienced a low prevalence of food insecurity [10]. The findings were consistent with other studies confirming that poorer households suffered from food insecurity most [12,13,15]. Therefore, to reduce food insecurity, in addition to the current income support for the elderly, the disabled, those with HIV/AIDS, and other vulnerable populations, governments should provide effective social protection to poorer households with children under 5 years old as they are vulnerable households.

Considering other factors, we also found that households in rural areas had lower food insecurity scores than households in urban areas. This can be explained by better access to homegrown food sources in rural households than in urban households, which rely more on purchasing food items. This result is explained by another study mentioning that in the rural areas, there is available space for gardening and access to food from neighbors [26].

### 4.1. Policy Implications

A few policies are generated by this study. First, this is the first time that the FIES has been integrated into the MICS in Thailand. This study demonstrates that FIES is a valid measure for assessing food insecurity in Thailand. It can be applied to estimate food insecurity prevalence in national representative household surveys as required by SDG 2.1.2.

Second, Thailand has yet to initiate a baseline prevalence of food insecurity as called for by SDG indicator 2.1.2. This can be done by integrating the FIES module into the National Health and Welfare Survey conducted biennially, or Socio-Economic Survey conducted annually by the NSO.

Third, despite a low level of food insecurity prevalence, the government should deliver its promises to SDG 2, ending hunger, achieving food security, improving nutrition, and promoting sustainable agriculture. Greater efforts should be given to end severe food insecurity (SDG indicator 2.1.2) and reduce the prevalence of under-nourishment (SDG indicator 2.1.1) in the light of increasing prevalence of stunting (SDG indicator 2.2.1), wasting, and obesity (SDG indicators 2.2.2) among children under 5 years old. All these efforts require evidence-based multi-sectoral actions for health guided by nationally representative household surveys using the FIES and MICS.

Finally, we recommend the government to introduce social protection measures such as food and nutrition subsidies or conditional cash transfer to poor households with children under 5 years old. There is strong evidence that a supplemental nutrition assistance program alone or in conjunction with education is an effective means of improving food security [27].

### 4.2. Limitation

There are a few limitations of this study. First, the study is not a nationally representative household survey as the sample frame covered households with children aged under 5 years old. Second, a cross-sectional survey nature of this study cannot establish causality of socio-economic factors on food insecurity. Third, other socio-economic status variables such as household income were not covered by this survey. Finally, the small sample size and the low prevalence of food insecurity did not allow analysis of the association between food insecurity and malnutrition.

## 5. Conclusions

This is the first study to determine the prevalence of food insecurity in households with children under 5 years old in Thailand. This study confirmed the validity of FIES in the Thai context. We found that the poorest households were more likely to suffer from food insecurity than the richest ones. We also found that households with children under 5 years old in rural areas suffered less severe food insecurity than households in urban areas. This suggests a need to improve social protection and provide income support to households with children under 5 years old, as food insecurity hampers child development and increases stunting and wasting. Moreover, social protection should be given to support poor households, particularly in urban areas, in ensuring food security in the light of the COVID-19 pandemic with increased unemployment and vulnerability.

## Figures and Tables

**Table 1 ijerph-19-05065-t001:** Infit and outfit test, measuring the validity of FIES among households with children under 5 years old in Thailand.

Questions of FIES	Infit	Outfit
Worried you would not have enough food to eat	1.01	2.01
Unable to eat healthy and nutritious food	1.21	1.42
Ate only a few kinds of foods	0.92	0.71
Skip a meal	1.05	2.27
Ate less than you thought	0.69	0.61
Ran out of food	0.80	0.13
Hungry but did not eat	1.06	2.73
Went without eating for a whole day	0.98	0.08

Normal range of infit = 0.7–1.3 and outfit < 2.0.

**Table 2 ijerph-19-05065-t002:** Characteristics of sample household and distribution of prevalence of food insecurity.

Interest Variables	Number of Sample Household (*n* = 861) (%)	Prevalence of Food Insecurity (%)
Overall		2.79
Household tertile *		
Poorest	287 (33.33)	6.62
Middle	287 (33.33)	1.05
Richest	287 (33.33)	0.70
Source of food		
Purchased	802 (93.15)	2.74
Homegrown	59 (6.85)	3.39
Geographic region *		
Central	173 (20.09)	3.47
North	340 (39.49)	0.29
Northeast	205 (23.81)	2.44
South	143 (16.61)	8.39
Place of residence		
Urban	258 (29.97)	2.33
Rural	603 (70.03)	2.99
Number of household members (Persons), ^a^ SD	4.80 (1.62)	
Number of household numbers		
≤3 people	190 (22.07)	2.63
>3 people	671 (77.93)	2.83
Receiving government’s support		
Not receiving	203 (23.58)	4.43
Receiving	658 (76.42)	2.28

^a^ SD = standard deviation; * chi-square test *p*-value < 0.05.

**Table 3 ijerph-19-05065-t003:** Association between socio-economic status and food insecurity among households: multiple logistic regression.

Variable of Interest	Adjusted OR (95% CI)	*p*-Value
Household tertile (ref. Poorest)		
Middle	0.11 (0.03, 0.37)	<0.001
Richest	0.07 (0.02, 0.34)	0.001
Source of food (ref. Purchased food		
Homegrown	1.28 (0.25, 6.54)	0.766
Geographical region (ref. Central)		
Northern	0.05 (0.01, 0.47)	0.008
Northeastern	0.48 (0.14, 1.71)	0.258
Southern	1.84 (0.64, 5.31)	0.259
Place of residence (ref. Urban)		
Rural	1.17 (0.41, 3.28)	0.770
Number of household members (Persons)	0.94 (0.71, 1.25)	0.683
Received government support (ref. No)		
Yes	0.52 (0.19, 1.38)	0.186

OR: Odds ratio. CI: Confidence interval.

**Table 4 ijerph-19-05065-t004:** Association between socio-economic status and food insecurity score using Tobit regression.

Interested Variables	Coefficient (95%CI) (*n* = 861)	*p-*Value
Household tertile (ref. Poorest)		
Middle	−1.94 (−2.72, −1.17)	<0.001
Richest	−3.61 (−4.63, −2.60)	<0.001
Source of food (ref. Purchased food)		
Homegrown	2.01 (0.91, 3.11)	<0.001
Geographical region (ref. Central)		
Northern	0.57 (−0.54, 1.69)	0.311
Northeastern	1.14 (−0.02, 2.29)	0.054
Southern	2.60 (1.41, 3.78)	<0.001
Place of residence (ref. Urban)		
Rural	−0.84 (−1.58, −0.10)	0.025
Number of household members (Persons)	0.05 (−0.16, 0.26)	0.641
Received government support (ref. No)		
Yes	−0.45 (−1.31, 0.41)	0.304
Constant	−4.04 (−6.05, −2.04)	

## Data Availability

Data and materials will be provided by the corresponding author upon request.

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
