# Peer review of "Thailand Prevalence and Profile of Food Insecurity in Households with under Five Years Children: Analysis of 2019 Multi-Cluster Indicator Survey"

_ijerph, 2022, doi:10.3390/ijerph19095065_

Round 1
Reviewer 1 Report
I found it interesting to read the manuscript. The manuscript is organized in a proper way. I am suggesting a few minor issues to improve the paper.
Line 12: under five-year-old children in Thailand.
Line 39: Spell out NCDs, (do the same for other abbreviations that used for the first time in the paper)
Line 66, and line 144: Need reference for the statement: food insecurity is less prevalent
Lines 112-118: Measurements: any pretest of the questionnaire?
Line 123: delete "
Line 207: any data on # of under five-year-old children? If yes, should include in the model. Since, the paper focuses on food insecurity in households with under five years children.
Power of the analysis to obtain results from 24 vs 837 households?
Thank you. Best wishes.
Reviewer 2 Report
I think the current articles are lack of innovation and need to be greatly improved before publication.
- The main conclution of this study is “The richest wealth households were less likely to suffer from food insecurity compared to the poorest households”.This conclusion seems common and not novel.
-
“The prevalence was lower than the 2016 FAO estimate of 4.8% for moderate or severe food insecurity in all Thai households.” Why households with under 5 year old child it has low prevalence than the average? Can you explain why?
- Why did you choose families with children under the age of 5 for research? According to your research, it seems that the food security situation of these families is higher than the overall level.
- Page 3 line 119-136, Although the main object of this article is families with children under the age of 5, these eight questions about food security are aimed at adults. Does this article focus on the food security situation of adults in families with children under the age of 5? This is puzzling.
- The conclusion and discussion are relatively simple and lack of in-depth discussion and analysis. Lack of innovation.
Round 2
Reviewer 2 Report
Major comment
Comment 1: I still miss some discussion on the novelty of the manuscript. The author repeatedly mentions the fact that there has been no study done to investigate the prevalence of food insecurity in Thailand. I would like to know some more on the issue.
Comment 2: The title of the manuscript should be further refined. In my opinion, the title does not present the objective of the manuscript very well.
Comment 3: Abstract: The section needs to be better worked out. In my opinion, it lacks sufficient clarity to be able to fully understand the manuscript.
The content of the methods in the abstract is not sufficiently clear. I would suggest improving it. The result needs to be improved too.
Comment 4: The literature review should be further refined. There are references in Thailand related to the issue (as stated in the paper). What are the findings? It would be good to have some information on the issue in order to put the study into perspective.
Comment 5: I also miss some justification on the fact that the focus of the study is on the household under five years of children. Although after I mentioned “these eight questions about food security are aimed at adults(Page 3 line 119-136).” last time, the author changed the content of the questionnaire from ‘adults’ to ‘individual’. But before that, the author has already carried out a questionnaire survey.
Comment 6: I would like to see a clearer justification for undertaking the study. The contribution of the paper to the literature needs to be better worked out.
Comment 7:. Materials and methods: The section needs to be better worked out. In my opinion, it lacks sufficient detail to be able to fully understand it.
Comment 8: Readability needs to be improved, especially at certain points. I would suggest writing in a more “concise” manner at certain points too.
Minor comments
- Lines 66-67: In my opinion, the sentence “Social protection which
Supports food security in poor households with children under five is recommended” needs to be somehow nuanced. - Lines 191-192: I would suggest inserting the reference number.
- Section 2.1: it is a bit confusing how the authors select the simple size. I would suggest being somewhat clear.
- Section 2.3.2: it is a bit confusing; the author mentioned multiple logistic regression and Tobit regression models but did not clear why select these models. I would suggest being somewhat clearer and describing these two models
- Lines 269-271: The authors used the infit test and outfit test to check the normality but didn’t clear about this test. I would suggest improving it.
